# Landslide Susceptibility Mapping Using Machine Learning: A Danish Case Study

**Angelina Ageenko, Lærke Christina Hansen, Kevin Lundholm Lyng, Lars Bodum *** and **Jamal Jokar Arsanjani**

Department of Planning, Aalborg University, Rendsburggade 14, 9000 Aalborg, Denmark;
aageen17@student.aau.dk (A.A.); lcha17@student.aau.dk (L.C.H.); kll17@student.aau.dk (K.L.L.);
jja@plan.aau.dk (J.J.A.)
* Correspondence: lbo@plan.aau.dk

**Abstract:** Mapping of landslides, conducted in 2021 by the Geological Survey of Denmark and Greenland (GEUS), revealed 3202 landslides in Denmark, indicating that they might pose a bigger problem than previously acknowledged. Moreover, the changing climate is assumed to have an impact on landslide occurrences in the future. The aim of this study is to conduct the first landslide susceptibility mapping (LSM) in Denmark, reducing the geographical bias existing in LSM studies, and to identify areas prone to landslides in the future following representative concentration pathway RCP8.5, based on a set of explanatory variables in an area of interest located around Vejle Fjord, Jutland, Denmark. A subset from the landslide inventory provided by GEUS is used as ground truth data. Three well-established machine learning (ML) algorithms—Random Forest, Support Vector Machine, and Logistic Regression—were trained to classify the data samples as landslide or non-landslide, treating the ML task as a binary classification and expressing the results in the form of a probability in order to produce susceptibility maps. The classification results were validated through the test data and through an external data set for an area located outside of the region of interest. While the high predictive performance varied slightly among the three models on the test data, the LR and SVM demonstrated inferior accuracy outside of the study area. The results show that the RF model has robustness and potential for applicability in landslide susceptibility mapping in low-lying landscapes of Denmark in the present. The conducted mapping can become a step forward towards planning for mitigative and protective measures in landslide-prone areas in Denmark, providing policy-makers with necessary decision support. However, the map of the future climate change scenario shows the reduction of the susceptible areas, raising the question of the choice of the climate models and variables in the analysis.

**Keywords:** predictive modelling; spatial prediction; Denmark; landslides; logistic regression; support vector machine; random forest; climate change; RCP8.5

## 1. Introduction

Landslides are traditionally perceived as a phenomenon limited to extremely steep and inhospitable areas [1]. However, this is not always the case, as many areas are affected by landslides [1]. Landslides are one of the most widespread geophysical hazards, which caused 378 disasters between 1997 and 2017, affecting approximately 4.8 million people and resulting in 18.414 fatalities. Furthermore, landslides have caused an estimated economic loss of USD 8 billion [2].

In Denmark, there has been little awareness of the risks associated with landslides. The country has to a limited extent been a part of international and European landslide databases,and only a few studies have been conducted on the subject until recently , mainly focusing on local field-based investigations of single events [3]. In 2015, the Geological Survey of Denmark and Greenland (GEUS) reported a total of 10 landslides, which is below the numbers in the rest of Europe, which ranges from 10 landslides (Denmark)

to 528.903 landslides (Italy) [4]. This, however, might not reflect the actual situation of landslides in Denmark compared to the rest of Europe, since each country has different landslide mapping strategies, and some countries have systematic landslide mapping, while other countries only report damaging landslides [4]. Orthophotos and high-resolution digital elevation models (DEMs) have allowed precise landslide mapping. The open data initiative in Denmark has granted wide access to actual DEMs of the entire country at a 40 cm spatial resolution [5], making it possible to conduct detailed and high-confidence mapping of landslides in Denmark, which resulted in a landslide inventory, which contains 3202 landslide polygons, indicating that landslides are more common than previously acknowledged [6].

Moreover, climate change is expected to contribute to more landslide occurrences in the future, as rising temperatures, heavy and more frequent precipitation, storm surges, and rising sea levels will influence bedrock stability [7]. Reference [8] provides a detailed list of climate-related changes and their respective effects on landslide response, including landslide-inducing factors such as the increase in total precipitation, rainfall intensity, temperature, wind speed, and duration, as well as alterations in the weather systems and the associated meteorological variability [8], which can play the role of preconditioning or triggering factors [9]. Despite the fact that many investigators have to some extent incorporated these climate variables in landslide susceptibility mapping as conditioning factors, e.g., [10–17], few research papers have extended their predictive models to consider climate evolution to make a quantitative assessment of future spatial variations of landslide susceptibility [18–23]. While there is a solid theoretical basis to assert increased landsliding activity as a result of projected climate changes [8], the impact of climate on landslide response at different spatial and temporal resolutions, however, remains unclear, requiring more studies conducted on the topic [24], which is central for understanding and predicting increasing or decreasing effects of these changes [25].

Denmark follows the global trends in temperature rise as projected in the IPCC scenarios, and by year 2100, it is expected that precipitation will increase by 25% in the winter, and storm surges, which happen statistically every 20 years, will happen every one or two years, while the mean sea level will rise at a rate of 2 mm/year [26]. These factors will likely have an accelerating impact on the frequency of landslides, requiring innovative approaches to map and predict current and future change in landslides [3].

### 1.1. Landslide Susceptibility Mapping

Landslide susceptibility mapping expresses the spatial probability of landslide occurrences. It is based on the main assumptions, synthesised by [24,27,28], that landslide events leave recognisable and identifiable traces [29–33], that they are more likely to happen in areas with similar conditions as earlier affected areas, and thereby, that the past can explain the future [34–36].

Several approaches for landslide susceptibility assessment are presented and reviewed in the literature [24,28,37–39]. Reference [39] point out two main approaches: direct and indirect. Direct approaches are generally more subjective and less reproducible, as these are based on experts' experience, decisions, and estimations, while the indirect approaches are considered to be more objective, as these are based on mathematical relationships [28,39]. Five broader method groups for landslide susceptibility mapping are distinguished in the literature, namely: geomorphological mapping, analysis of landslide inventories, heuristic approaches, process-based methods, and statistically based modelling [24,34]. While the quality of landslide susceptibility assessment provided by the first three groups can suffer from the incompleteness and the quality of the used inventories, the complexity of the area of interest, and the ability and judgements of expert investigators [40–42], process- and statistically based approaches, which provide physically based models and analysis of the relationships between causative factors and landslide occurrences are considered to be favoured quantitative methods in landslide susceptibility studies with a shift towards the machine learning approach in the last decade [24]. Reference [37] divided methods for

landslide susceptibility assessment into knowledge-driven, data-driven, and physically based. The data-driven methods such as machine learning show promising results and have become common for landslide susceptibility modelling over larger regions [43–45], which might be due to insufficient and limited geotechnical data used in physically based models, their complexity, and the associated time consumption [46–48]. Data-driven modelling is performed under the main assumption that landslides are likely to happen in similar conditions of the past and present landslide events. The precise relationship between landslide presence/absence and these conditions is not always well known, as these conditions are hard to measure in larger regions, which is why they are represented by a number of predictors, or independent variables [48]. The advantages of data-driven machine learning algorithms over statistically based methods are higher accuracy without the need for a large amount of historical landslide data [49], the absence of the requirement for certain statistical assumptions such as a normal distribution and a priori linear relationships, and usefulness in the determination of novel relationships within the dataset [45]. Some of the well-established and well-performing machine learning algorithms within landslide susceptibility mapping have until now been logistic regression [50], support vector machines [51], decision trees, and ensemble methods such as bagging, random forest, and rotation forest [52–55], with more advanced hybrid and deep learning techniques emerging in the literature [56,57]. There is no consensus on the optimal machine learning technique for landslide susceptibility modelling [48,58]; therefore, the performance of different models normally needs assessment and comparison in different cases [59].

In terms of case diversity, data-driven landslide susceptibility studies are predominantly concentrated on hilly or mountainous areas with little or no attention given to flat low-lying terrains [43–45,47,49,53,59–61], with China, Italy, India, and Turkey being the most represented countries, highlighting the needed effort to reduce the clear geographical bias by investigating new regions of interest [24].

### 1.2. Problem Statement and Study Objectives

The data-driven approach based on machine learning algorithms has proven to perform well for landslide susceptibility mapping in the mountainous and hilly study areas. However, the literature asks for further advancement in terms of diversity of the case studies. Moreover, the impact of climate change using the future climate variables on landslide susceptibility remains unclear. Hence, the main objective of this study is to set up the machine-learning-based toolkit for mapping landslide susceptibility in a low-lying landscape with a relatively large number of landslide events and to project the susceptibility into the future RCP8.5 scenario. Given this main objective, this study aims at answering the following questions:

1. How well can well-established machine learning algorithms be employed for landslide susceptibility mapping given the low-lying flat landscape of Denmark?
2. How are the various variables related to the landslide presence locations? What is the importance of the different variables in the prediction model?
3. Can the impact of changing climate on landslide susceptibility be modelled for the future climate scenario?

## 2. Data and Materials

### 2.1. Area of Interest

The chosen area of interest (AOI), shown in Figure 1, is located in the eastern part of Jutland, Denmark, between Horsens and Kolding, and is 972 km$^2$, containing 453 historical landslides, covering an area of 3.33 km$^2$ in total. The area of interest was chosen due to the high concentration of landslides, where three different types of landslides are represented: slides (394), flows (31), and spreads (28). This includes 189 inland landslides and 264 coastal landslides, depending on their distance from the coast, where the inland landslides are defined as being farther than 300 m away from the coast. The area of the individual landslides varies between 96 m$^2$ for the smallest and 73,429 m$^2$ for the largest.

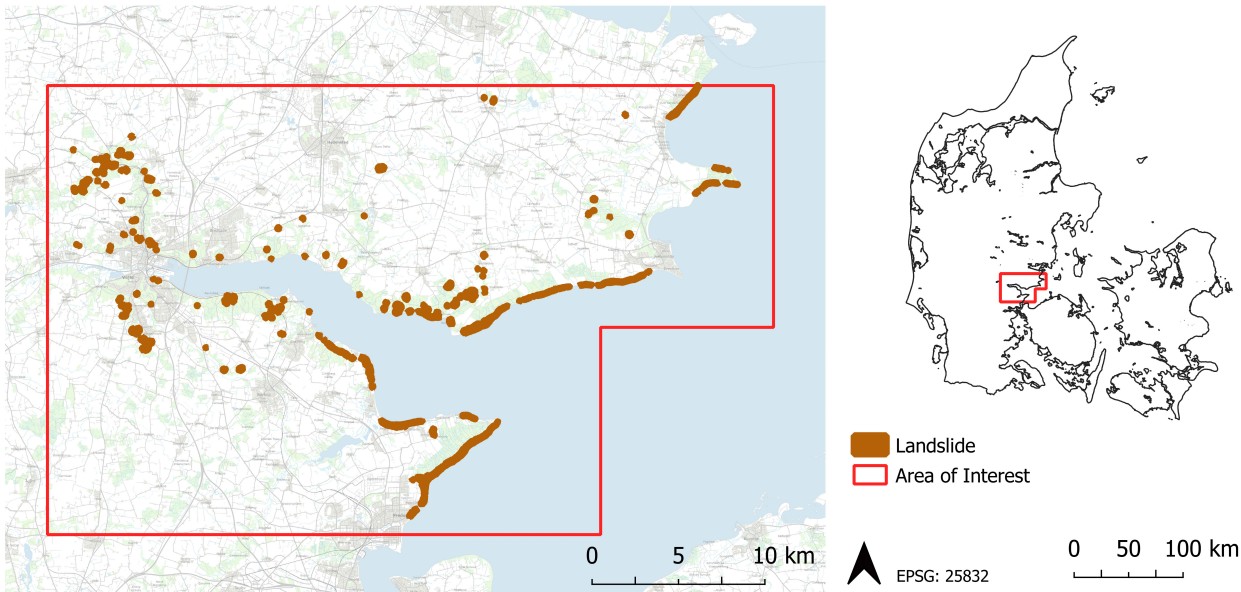

**Figure 1.** The selected study area.

## 2.2. Landslide Inventory

The landslide inventory used in this study is produced by GEUS based on the expert-based interpretation of the DEM from 2015 and multitemporal high-resolution orthophotos provided by the Danish Agency for Data Supply and Efficiency (SDFE) [6]. The multidirectional hillshade derived from the DEM was used to identify landslides based on clearly visible scarps and/or displaced units, and this process was supported by the time series of orthophotos [6]. This resulted in a high confidence map of landslides, where the landslide polygons were produced by drawing vertices around the identified landslides, only including landslides with an area above 25 m$^2$ [6]. The inventory does not contain any information on when the mapped landslides occurred or what the current state of their activity is. The subset of the national landslide inventory used as a dependant variable in this study comprises 453 landslides. Centroids were computed for each landslide, giving a sample size of 453 landslide presence points in the AOI, seen in Figure 2. Generating only one point for each landslide ensures equal treatment for all the landslides regardless of their size [62]. Creating points that represent non-landslides, or the absence of landslides, is necessary in order for ML models to distinguish between landslide and non-landslide classes [63]. A random sample of non-landslide points, equal to the amount of the landslide centroids, was generated with the restrictions for the water bodies and the landslide occurrence polygons with an outward buffer of 50 m. With an equal number of landslide and non-landslide samples, one can directly interpret the unadjusted predicted probabilities [62] and avoid a class imbalance and model bias, where areas prone to landslides might risk being classified as safe because of the overrepresentation of non-landslide samples [63].

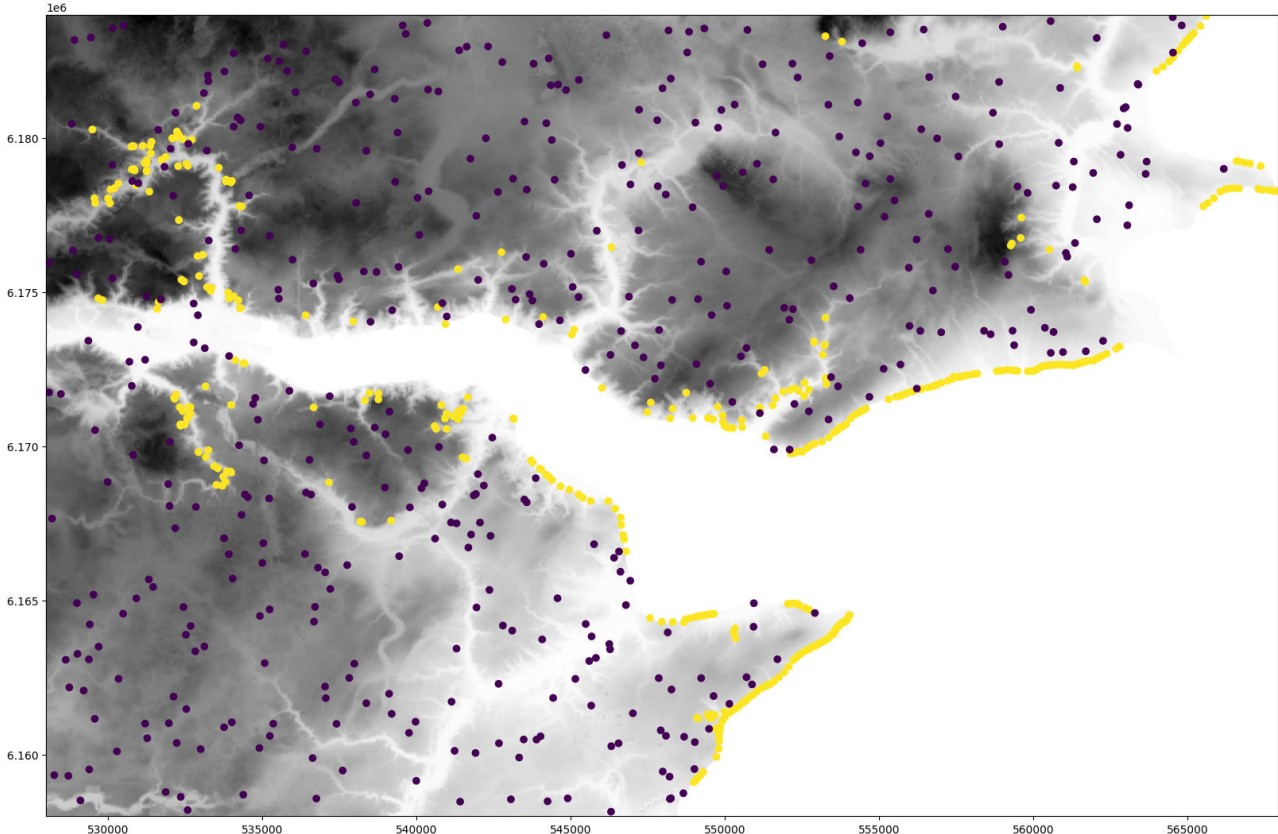

**Figure 2.** Overview of the centroid points of landslides (yellow) and an equivalent amount of randomly sampled non-landslide points (purple).

*2.3. Predictive Variables*

In addition to the dependent variable, susceptibility assessment requires information about factors that can explain landslide occurrences/absence. Predictive variables are landslide predisposing factors used as the input for the prediction. It is crucial to choose only relevant and suitable factors that cause landslides, as redundant predictors may lead to unnecessary noise, lowering the predictive capability of the machine learning models [63]. A set of conditioning factors should be chosen with respect to a particular area and should normally take into account the study area characteristics, the type of landslides, the scale on which analysis is conducted, and the availability of the data [64]. There is no universal standard or guidelines as to which factors to select, but there are several common internal and external predictors used in previous landslide susceptibility assessment studies [63]. The internal factors include, but are not limited to elevation, profile curvature, slope, plan curvature, distance from faults, aspect, distance from rivers, land form, and lithology, while the external factors include precipitation, distance from roads, and seismic activity [63].

This study considers a total of 29 variables, including topographic, hydrological, geomorphological, anthropogenic, and climatic variables, seen in Table 1. DEMs are among the most crucial data products used in landslide modelling because of a number of explanatory terrain attributes that can be derived from them [62]. These DEM derivatives serve as proxies for surface processes and different geophysical conditions and are used to represent complex geomorphological relationships and to make them more simple [48].

Elevation is always used in landslide susceptibility studies, as it functions as a proxy for variability in rainfall, vegetation, and soils [62]. The original DEM in this study was resampled to 2 m, as this resolution helps to reduce the size of the DEM for computations, while keeping the resolution high enough to represent the landslides, where the smallest one has an area of around 11 × 8 m.

**Table 1.** Overview of the predictive variables.

| Category | Variable | Type | Spatial Resolution | Source |
|---|---|---|---|---|
| Topography | Elevation | Continuous | 0.4 m | [65] |
| | Slope | Continuous | 2 m | - |
| | Aspect | Continuous | 2 m | - |
| | Planform curvature | Continuous | 2 m | - |
| | Profile curvature | Continuous | 2 m | - |
| | TPI | Continuous | 2 m | - |
| | TRI | Continuous | 2 m | - |
| | Roughness | Continuous | 2 m | - |
| | Slope std | Continuous | 2 m | - |
| Hydrology | SPI | Continuous | 2 m | - |
| | TWI | Continuous | 2 m | - |
| | Distance from streams | Continuous | 2 m | |
| | Distance from coast | Continuous | 2 m | - |
| | Depth to ground water | Continuous | 100 m | [66] |
| Geomorphology | Landscape types | Categorical | 1:200,000 | [67] |
| Geology | Topography of the pre-Quaternary surface | Categorical | 1:250,000 | [68] |
| | Pre-Quaternary deposits | Categorical | 1:50,000 | [69] |
| | Surface geology— soil types | Categorical | 1:25,000 | [70] |
| Anthropogenic | Distance from roads | Continuous | 2 m | - |
| | Distance from railroads | Continuous | 2 m | - |
| | Distance from quarries | Continuous | 2 m | - |
| Climate | Mean temperature | Continuous | 1 km | |
| | Mean wind | Continuous | 1 km | |
| | Max daily precipitation | Continuous | 1 km | |
| | Max 14-day precipitation | Continuous | 1 km | [71] |
| | 5-year extreme occurrence of precipitation | Continuous | 1 km | |
| | 50-year extreme occurrence of precipitation | Continuous | 1 km | |
| | Cloudburst | Continuous | 1 km | |

Slope is included, as it controls the retaining and destabilising forces impacting a slope as a larger resistance is required to keep a steep slope stable than to keep a gentle slope stable [24]. Slope is decomposed into two elements: gradient and aspect. Gradient indicates maximum change in altitude, while aspect is defined as the compass direction of this change [72]. Aspect is a circular parameter, which is decomposed into its sine and cosine elements describing the terrain's exposedness to the east and to the north, respectively, to avoid discontinuity [62]. Planform and profile curvature are used as predictor variables, as profile curvature can be used as a measure for the flow acceleration or deceleration down the slope, while plan curvature indicates the flow convergence or divergence down the slope [72]. The Topographic Wetness Index (TWI) and Topographic Position Index (TPI) are included, as they are often regarded as proxies for the spatial soil variability. The TWI affects soil moisture in valleys, while the TPI expresses the geomorphological setting in a numerical way [62], where negative values represent the areas that are located lower than the surroundings such as valleys, and the positive values indicate the areas located higher than the average surroundings. While the other DEM derivatives and indices are the output of the terrain processing tools, the TWI is calculated according to [72]:

$$TWI = ln\left(\frac{a}{tan(\beta)}\right) \tag{1}$$

where *a* is the contributing upslope area (catchment area) and *β* is the slope angle in radians

The Stream Power Index (SPI) is used to measure the erosive power of a flow of water [72]. The SPI is computed according to the formula:

$$SPI = a * tan(\beta) \tag{2}$$

where *a* is the contributing upslope area (catchment area) and *β* is the slope angle in radians. The SPI is further log-transformed. Other variables may also be effective in landslide susceptibility modelling such as the standard deviation of elevation or slope. These express terrain roughness, which is expected to be smaller in stable areas than in landslide areas [24]. For this study, the following variables that describe terrain roughness are selected:

The Terrain Ruggedness Index (TRI) is regarded as a measurement of the land surface condition. The TRI assists in describing terrain as smooth or rugged, and it depicts the local variance of curvatures and gradients [72]. Roughness is another variable used to describe the surface condition [73]. The standard deviation (std) of the slope was also suggested in [48] as a measure for surface roughness. It is computed as the standard deviation of the slope values in a 3 × 3 cell neighbourhood.

Distance from streams and distance from coastline are included as predictor variables. The rationale is that these distances can capture certain conditions that might contribute to making the bottom of a slope unstable. Only the larger streams (above 2.5 m in width) are included here as proposed by [24,74].

Distance from roads, railways, and quarries are included as explanatory variables, as areas in proximity to these are highly affected by human activities during establishment, use, and maintenance, and this human interference might affect the stability of slopes [1]. The variables with distances from streams, coastline, roads, railways, and quarries are computed using the euclidean distance from the respective features. Only distances up to a certain threshold are considered, and thus, the layers have been trimmed and reclassified accordingly. As an example, it was assumed that streams 500 m away have an equal influence on landslide occurrences as streams that are 300 m away. The threshold for each of the feature variables is seen in Table 2.

As proposed by [60], a distance threshold of 100 m was chosen for the roads and railways, as distances exceeding this threshold were assumed not to have any impact on the landslide occurrences. Distances exceeding the thresholds were assigned the same value as the threshold.

**Table 2.** Distance threshold for the predictor features.

| Feature | Threshold (m) |
| :---: | :---: |
| Streams | 300 |
| Coastline | 300 |
| Roads | 100 |
| Railways | 100 |
| Quarries | 250 |

The soil type dataset describes the geology of the surface [70] and the geomorphology related to the formation processes in the landscape [67]. The map of the underground describes bedrock geology and depicts the stratigraphy of the AOI [69], and the pre-Quaternary topography describes the elevation of the pre-Quaternary surface [68]. The distance from faults is not included as a predictive variable, due to their limited presence in the AOI.

*2.4. Climate Data—Present and Future*

The climate data for Denmark are produced and computed by DMI on the basis of up to 57 climate model results and represent both data of the current climate, as well as data

for the future climate [26]. The data consist of 18 climate-related variables; however, only those that might have relevance to the occurrence of landslides were included.

The current climate data are based on the average of the reference period 1981–2010. This period was chosen as there are no timestamps in the landslide inventory. Additionally, the DMI's data for future climate change are classified into three time periods: 2011–2040, 2041–2070, and 2071–2100, where the period from 2071–2100 was chosen. The climate data set represents future climate change in accordance with the IPCC scenarios RCP4.5 and RCP8.5. For the susceptibility mapping of landslides in this study, the data for the RCP8.5 scenario were chosen, since DMI recommends using the RCP8.5 scenario for projects with a planning horizon past year 2050, where the robustness of the model for climate change adaptation is required [75]. Since the data for future climate change are computations derived from climate models, they are associated with uncertainties. These uncertainties are expressed in the data as the median and the 10th and 90th percentiles, where the median provides the best estimate [26], and was therefore chosen in this study, as seen in Table 3. The values are the relative change in percent, between the reference period of 1981–2010 and of 2071–2100 according to the RCP8.5 scenario [75].

**Table 3.** Overview of the chosen climatic variables, their units, and the relative change in percent between 1981–2010 and 2071–2100 for the whole of Denmark.

| Variable | Unit | Relative Change (%) |
|---|---|---|
| Mean temperature | Degrees C | 3.37 |
| Mean wind | m/s | −0.66 |
| Mean precipitation | mm/day | 13.75 |
| Max daily precipitation | mm/day | 23.02 |
| Max 14-day precipitation | mm/14 day | 15.39 |
| 5-year extreme occurrence of precipitation | mm/day | 19.37 |
| 50-year extreme occurrence of precipitation | mm/day | 23.83 |
| Cloudburst | Number of yearly occurrences | 69.00 |

The vertical distance from groundwater was included as it is assumed to influence slope stability as saturated soil makes it more unstable and thereby more prone to sliding. The data set shows the average vertical distance from the ground surface to the surface of the groundwater for the reference years 1990–2020. The future RCP8.5 scenario includes the average expected changes in the ground water level in the period 2070–2100.

The common preprocessing step for the above-mentioned variables is to prepare them as raster layers with the common extent, resolution, and projection to the coordinate system ESPG 25832 to create a raster stack. All the variables were clipped to remove the cells representing water, as they are irrelevant in the landslide analysis. Since the DEM is crucial for landslide susceptibility assessment, all other raster layers were resampled to its resolution of 2 m. The categorical variables binarised using a one-hot encoding scheme that produces n-1 new binary variables for every categorical dataset.

## 3. Methods

The methodological framework for this study is shown in Figure 3 and includes several steps starting with the definition of the problem for machine learning as a binary classification problem (landslide/non-landslide), followed by data selection and acquisition, and the preparation of the variables. Then, the machine learning models are set up, assessed, and validated. The models and the final susceptibility maps are compared, and the map for the future landslide susceptibility is produced.

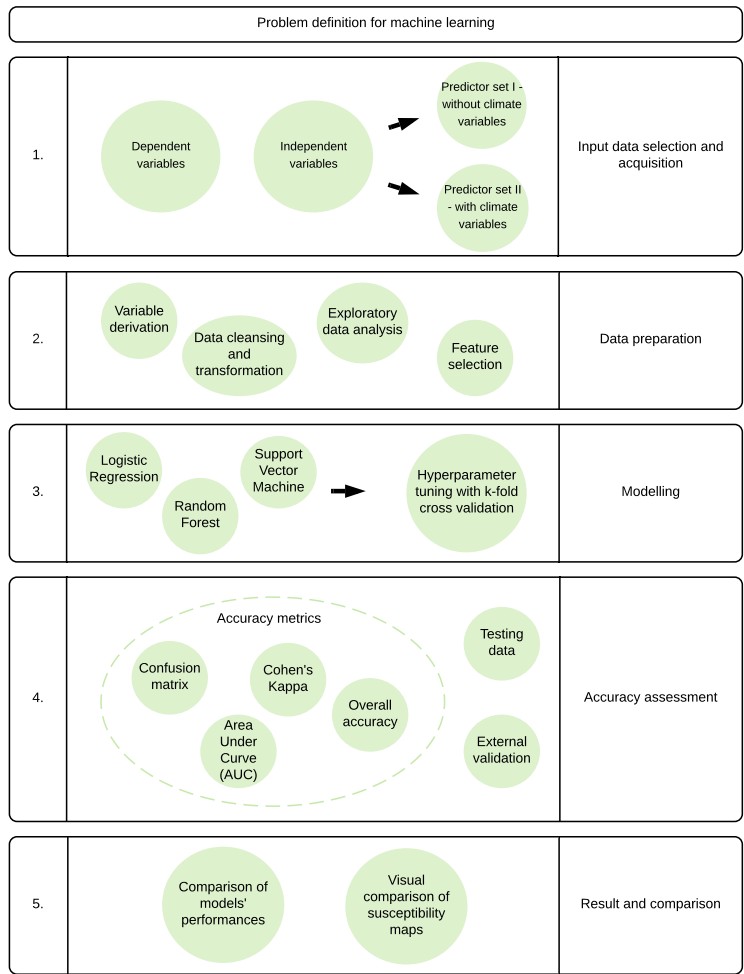

**Figure 3.** Methodological framework for the problem solution.

To investigate whether climatic variables can be used to predict the impact of climate change on landslide susceptibility in Denmark, two predictor sets were made: Predictor Set I—without climate variables and Predictor Set II—with climate variables, as seen in Figure 3. The methodological framework is the same for both predictor sets, and the models with each predictor set were run in a parallel manner. Additionally, a prediction for the future RCP8.5 scenario for years 2071–2100 was made for Predictor Set II, based on the projected future evolution of the chosen climate variables.

### 3.1. Feature Selection

In this study, a principal component analysis (PCA) was performed with the purpose of understanding the variables and the relationship between them. The PCs are a simpler representation of the data, and they can often identify underlying characteristics in the data, which assists in the feature-engineering process [76]. Since the variables do not have the same units and scale, all the variables were first standardised, i.e., centred around 0 and scaled to achieve the standard deviation of 1, to make them comparable.

The PCA resulted in 26 principal components, which describe 100% of the variability in the data. The first principal component captures around 25% of the variability in the data set. The second and third component capture 15% and 10%, respectively, and the first 16 principal components are enough to cover 95% of the variability in the data. The first component is influenced significantly by the climatic variables such as rain, wind, and temperature, and the second component is influenced mainly by the DEM derivatives such as slope, TRI, and roughness.

The loadings are plotted with the PC1 as the x-axis and the PC2 as the y-axis, as seen in Figure 4. The loadings plot visualizes how strongly each variable influences the principal component, where the angle between the vectors indicates correlation between the variables, where a small angle indicates a high positive correlation and an angle of 180 indicates a negative correlation. Based on the plot, it is clearly visible that the climatic variables concerning rain—rain_max14_day_ref, rain_average_ref, rain_5_year_ref, rain_50_year_ref—are heavily correlated. Furthermore, slope_degrees, roughness, and TRI express a strong positive correlation.

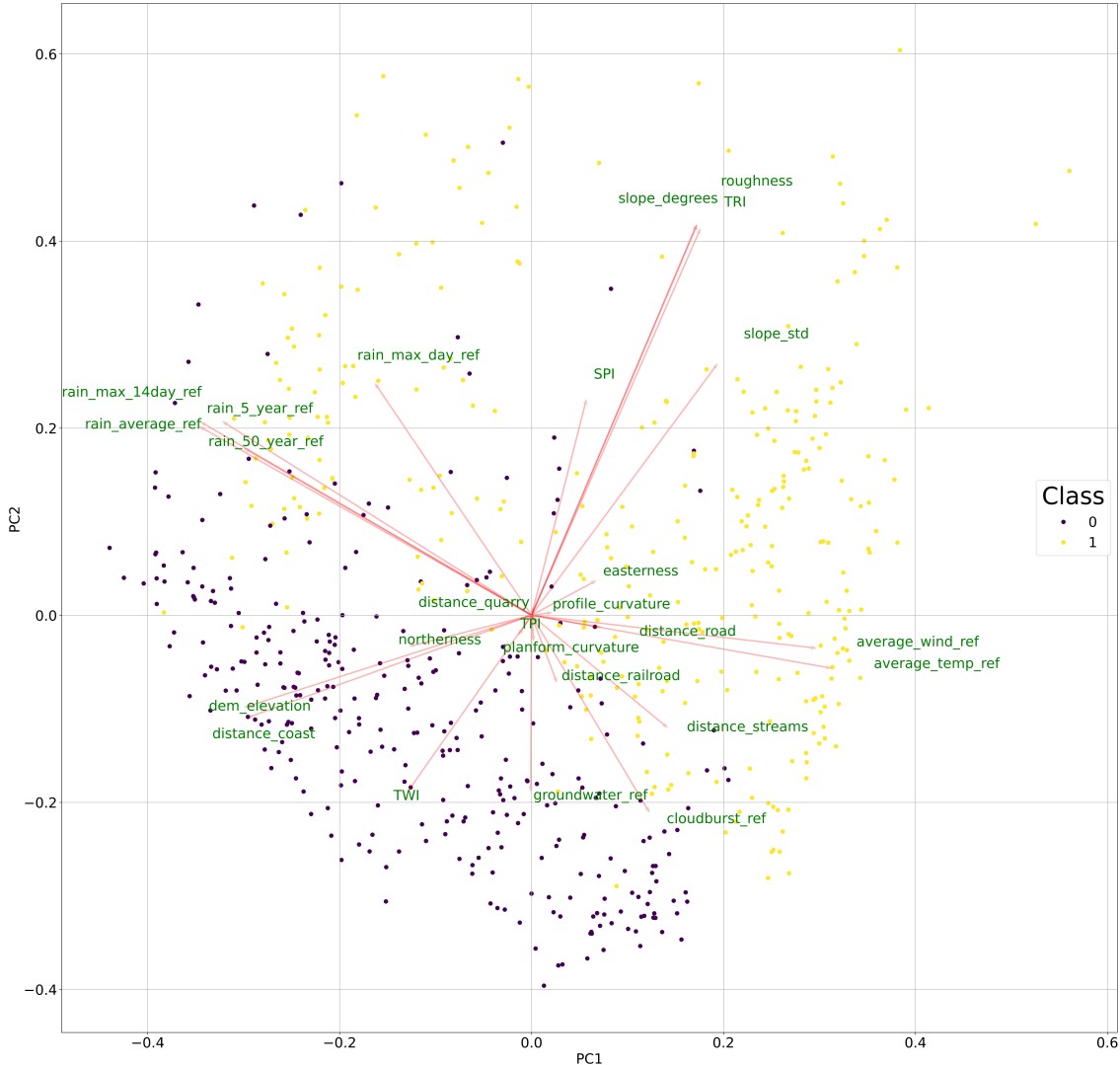

**Figure 4.** PCA biplot with principal component loadings of variables, where class 0 indicates non-landslide samples, while class 1 is landslides.

To investigate the data further for correlated variables in order to avoid redundant information, a correlation matrix was computed, as seen in Figure 5.

It is noticeable that the following variables concerning rain—rain_max14_day_ref, rain_average_ref, rain_5_year_ref, rain_50_year_ref—are strongly correlated, with correlation coefficients between 0.86 and 1. Furthermore, a strong correlation was observed between several of the DEM derivatives such as slope_degrees, roughness, and TRI, with values between 0.99 and 1, which supports the observations made from the PCA. The strongly correlated variables should be removed in order to avoid redundant variables that might lead to unwanted complexity in the models and degrade their performance [76]. On this

basis, the following variables exceeding the pairwise correlation threshold of $\pm 0.75$ were removed from the data, as they were considered to be substantially correlated:

- Slope degrees;
- Roughness;
- planform_curvature;
- profile_curvature;
- average_wind_ref;
- rain_max_14day_ref;
- rain_5_year_ref;
- rain_50_year_ref.

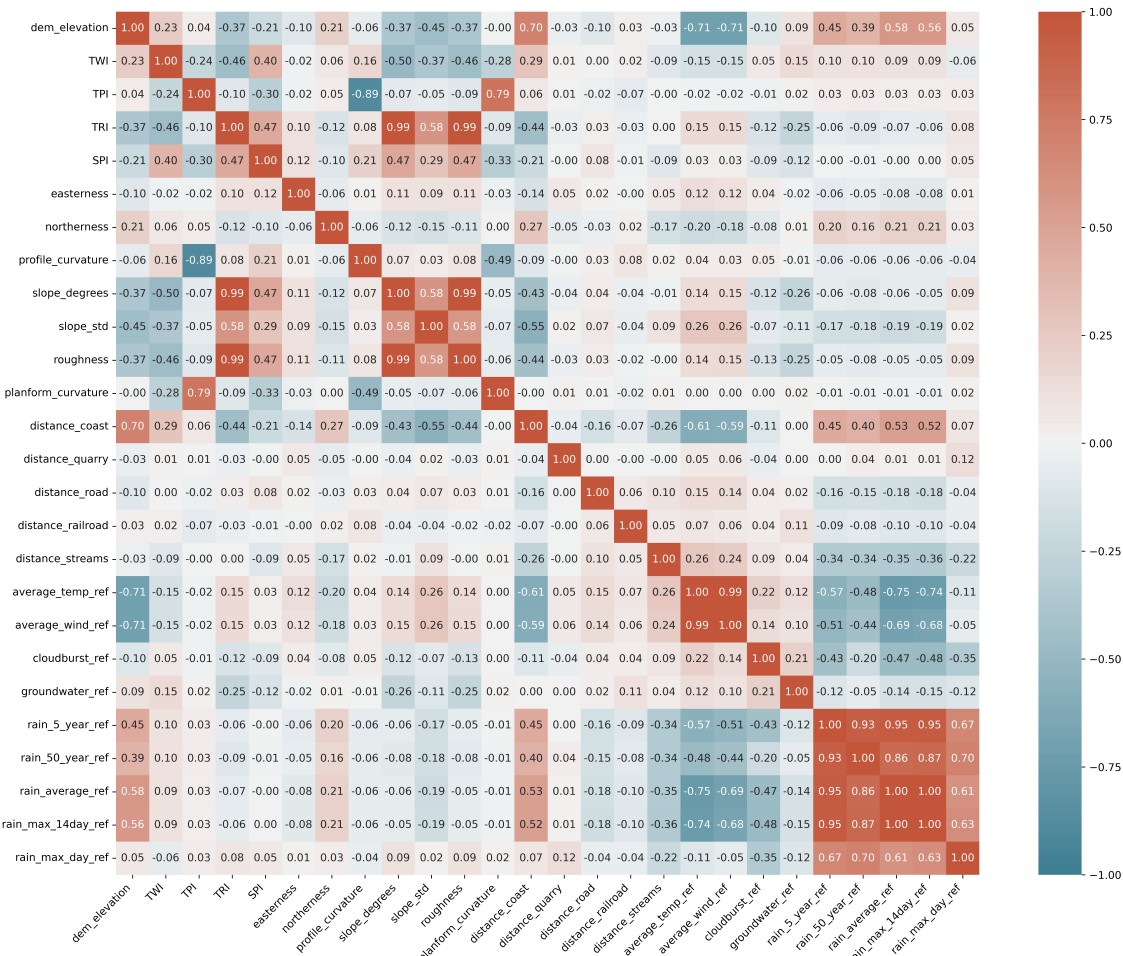

**Figure 5.** Matrix with the Pearson correlation between the predictive variables.

To determine the importance of the variables, the random forest model was used, as it is fairly simple and has a feature importance function [77]. From the RF feature importance, seen in Figure 6, it is noted that distance_quarry has no importance for the model, while distance_railroad and distance_road have a minor impact on the model. On this basis, they were regarded as insignificant for the prediction model and were removed. The variables that were the most important were TRI, the standard deviation of slope, distance from the coast, elevation, TWI, and SPI, which are mainly derivatives of the DEM.

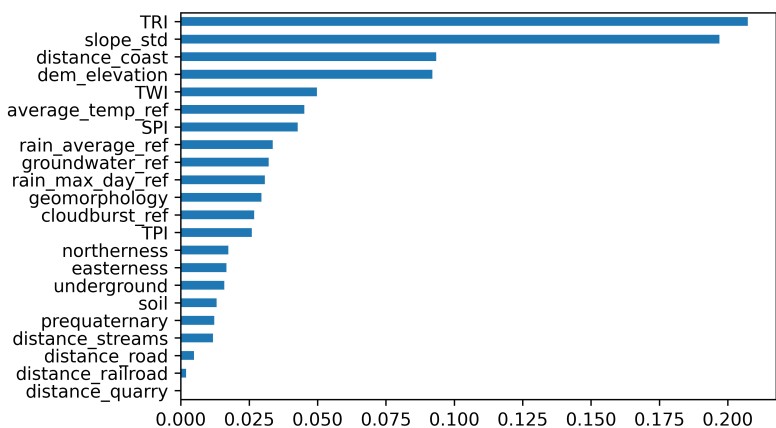

**Figure 6.** Feature importance from the RF model.

After having examined the correlation matrix and the feature importance graph, the final predictor sets were selected and can be seen in Table 4.

**Table 4.** Overview of the final selection of the variables for the two predictor sets.

| Predictor Set I | Predictor Set II |
| --- | --- |
| dem_elevation | dem_elevation |
| slope_std | slope_std |
| TWI | TWI |
| TPI | TPI |
| SPI | SPI |
| TRI | TRI |
| easterness | easterness |
| northerness | northerness |
| distance_coast | distance_coast |
| distance_streams | distance_streams |
| geomorphology | geomorphology |
| soil | soil |
| prequaternary | prequaternary |
| underground | underground |
| | average_temp |
| | rain_average |
| | rain_max_day |
| | groundwater |
| | cloudburst |

*3.2. Landslide Susceptibility Modelling: Set up and Tuning*

3.2.1. Random Forest

The random forest model is a supervised learning ensemble classifier that involves more than one decision tree to make a "decision forest". Each of the individual decision trees vote on what they think the outcome is, and the result of the random forest is the outcome with the majority of votes [78]. The collective time it takes to train the RF is less than other supervised classification methods, which makes it a viable choice. The RF was selected in this study, as its predictive performance has been shown to be one of the most promising for landslide susceptibility mapping [48,61].

3.2.2. Support Vector Machine

SVM is a well-established algorithm within machine learning and works well in practice across numerous applications. SVM is a supervised learning technique that learns from training data and tries to predict and generalise the data by separating the data based on hyperplanes, which splits the data into categories. The hyperplane found by SVM

is estimated in such a manner that the distance (margin) to the nearest training data is maximised on either side of the hyperplane [76]. SVM was selected as its kernel tricks provide a unique solution for complex problems, though the parameter selection can be intensive in terms of computation [62].

### 3.2.3. Logistic Regression

Logistic regression is a classification algorithm that is commonly used when the data in question are binary, meaning that they belong to either one class or the other. Logistic regression is in theory a linear regression; however, it uses a more complex cost function defined as the "sigmoid function/logistic function". LR is considered to be one of the most frequently used algorithms in landslide susceptibility modelling on the regional scale [62,79].

The construction of the ML models was performed using an open-source Python library, Scikit-Learn [80]. The data used for the models were split into training and testing subsets in order to make an unbiased assessment of the models' performance. The training data were used to fit and develop the models, and the testing data were used to validate and test the trained model. There is no uniform guideline for how the data should be split [81]. In this study, the data set was partitioned into a training subset consisting of 70% of the data and a testing subset of 30%. The split was stratified after classes, in order to achieve a balanced amount of landslide and non-landslide samples in both the training and test sets. The random state was set to ensure the reproducibility of the models.

### 3.3. Hyperparameter Tuning and K-Fold Validation

Tuning of the hyperparameters in this study was performed using Grid Search. With Grid Search, a candidate hyperparameter or a set of candidate hyperparameters is chosen, and models are built for each possible combination of the chosen values. Ten-fold cross-validation was used to evaluate each distinct parameter value combination, to analyse how well each candidate performed. Usually, a k-value of 5 or 10 is used in machine learning, but there is no formal rule as to how many it should be. It is worth mentioning however that a higher k is more computationally burdensome [76].

When the results of the 10-fold validation have been computed, the most optimal hyperparameter combination was chosen for the final model to fit to the training data on the basis of the empirically best results [81]. The hyperparameters resulting from the Grid Search and used in this study are seen in Table 5.

**Table 5.** Hyperparameters used in the ML models.

| Model | Parameter | Predictor Set I | Predictor Set II |
|---|---|---|---|
| RF | Number of estimators | 100 | 200 |
| | Max_features | "auto" | "log2" |
| SVM | C | 10 | 1 |
| | Gamma | "auto" | 0.1 |
| | Kernel | "rbf" | "rbf" |
| LR | C | 1 | 1 |
| | Penalty | "l1" | "l1" |
| | Solver | "liblinear" | "liblinear" |

The number of estimators in RF determines how many "votes" every classification has to complete to make the final decision, while max_features is the number of features randomly selected at the node split. The four common kernel functions in SVM are: "linear", "polynomial", "radial basis function", and "sigmoid", of which the radial basis function (RBF) was chosen for both predictor sets, indicating the non-linearity of the datasets. This function requires two parameters, Gamma and penalty (C). Gamma is a kernel coefficient that defines the influence of a sample, where larger gamma values mean that the influence of that data point does not spread far. The penalty (C) defines the desire

to avoid misclassifications. A large penalty forces SVM to choose a hyperplane with a narrower margin, if this hyperplane helps to classify more points correctly. On the contrary, a smaller penalty makes SVM prefer a hyperplane with a wider margin, even if it means that this will label more points wrongly. The "solver" hyperparameter for LR determines which algorithm to use. The solver used for both predictor sets is "liblinear", which is efficient with smaller datasets. The norm, used for penalisation, is l1, which might limit the size of the coefficients, while C is, similarly to SVM, a regularisation hyperparameter.

*3.4. Accuracy Assessment*

Having performed a classification task, the accuracy assessment of the results was conducted to ensure the quality of the classification and evaluate its performance. One of the methods of conducting accuracy assessments used in this study is producing a confusion matrix in order to compare the ground truth with the results from the classification [82]. In the study case, a false negative (FN) means that a model has failed to detect an actual landslide, while a false positive (FP) indicates that a non-landslide has been falsely labelled as a landslide by the model. FNs are considered as the "worst-case scenario", since some potential landslide-prone areas will be overseen and will be shown as "safe" on a susceptibility map. This consideration is taken into account when the model evaluation is performed. From the confusion matrices, a number of descriptive measures of the classification such as overall accuracy and Cohen's Kappa are computed. Overall accuracy indicates the proportion of all the reference samples that were classified correctly [82]:

$$Overall\,Accuracy = \frac{TruePositives + TrueNegatives}{TotalSample} \qquad (3)$$

Cohen's Kappa statistic is a measure of the extent of which the percentage of correct values in an error matrix is due to "true" agreement or "chance" agreement, as even a completely random assignment of classes will produce a percentage of correct values [82]:

$$Kappa = \frac{OA - CA}{1 - CA} \qquad (4)$$

where *OA* is the overall accuracy of the model and *CA* is the chance agreement.

The ROC curve measures the performance of classification problems at different threshold settings. The ROC is a probability curve for binary classification problems, and the AUC indicates how well a model can distinguish between classes. The higher the AUC value, the better the model is at predicting the correct classes [83]. In this study, the ROC and AUC were used to compare the performance of the classification algorithms. The ROC curve is plotted with the true positive rate (TPR) against the false positive rate (FPR), where the TPR is on the x-axis and the FPR is on the y-axis. The TPR, also called sensitivity, corresponds to the proportion of positive samples that are correctly classified as positives (true positives) in relation to all actual positives (true positives and false negatives) [83]:

$$Sensitivity = \frac{TruePositive}{TruePositive + FalseNegative} \qquad (5)$$

The true negative rate (TNR) or specificity is the classifier's ability to correctly classify negative samples as negatives. In contrast to sensitivity, specificity corresponds to the proportion of negative samples that are correctly classified as negative in relation to all actual negatives (true negatives and false positives) [83]:

$$Specificity = \frac{TrueNegative}{TrueNegative + FalsePositive} \qquad (6)$$

The *FPR* used to plot the ROC curve is computed by:

$$FPR = 1 - Specificity \qquad (7)$$

*3.5. External Validation*

When selecting model hyperparameters, there is a risk that a model overfits by over-learning the relationship between the explanatory variables and the classes in the training data. This causes a problem, since over-interpretation of patterns in the training set means that the predictive power of the model degrades when exposed to new data [81]. To avoid that, the model with the tuned hyperparameters is validated on an external data set, which has not been used in the training of the model and the tuning of the hyperparameters. A data set from the outside of the AOI is used for this purpose. The area for external validation is located in the southern part of Denmark, from Haderslev to the Danish–German border including the island of Als; see Figure 7. This area was chosen based on the available landslide data provided by GEUS and because of the high concentration of landslides in the area. For the landslide and non-landslide samples in the external validation area, the centroid sampling strategy was used, as previously described, which resulted in 260 landslide and 268 non-landslide points.

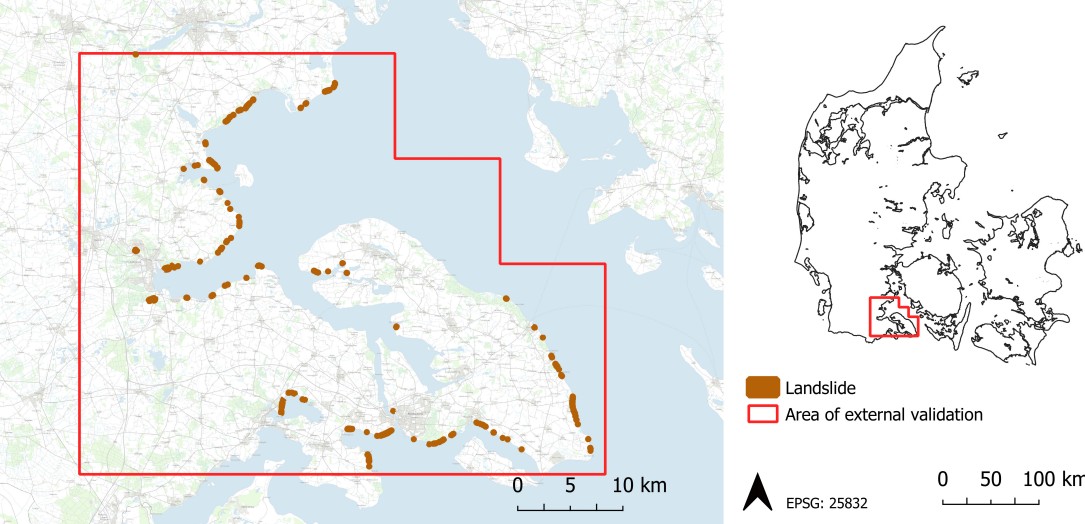

**Figure 7.** Area for external validation, excluding the island of Soenderby.

## 4. Results

The accuracy metrics for both predictor sets is seen in Table 6. For Predictor Set I, the overall accuracy for the three algorithms was nearly equal at 0.91 and 0.92. The Kappa value of SVM was slightly higher than the values of RF and LR, though all three models have a Kappa value above 0.82, which indicates almost perfect agreement, meaning that chance is not the reason for the high accuracy. The sensitivity of the three models with Predictor Set I are generally represented by equally high values, and the specificity values range from 0.88 to 0.92. The overall accuracy of the validation on the external area indicates that the SVM and LR models do not generalise well on the unseen data. This is attributed to the different spatial distribution of the pre-Quaternary topography classes. Classes 8 and 9 are not represented in the external validation area, while the presence of class 7 is limited, compared to the AOI. Classes 5 and 6 are characterized by overweighting of the landslide points in the AOI, while these classes in the external validation area contain predominantly non-landslide samples. The result emphasises the robustness of the RF algorithm to handle the unknown and missing data.

**Table 6.** Accuracy metrics of all models with both predictor sets.

| | Overall Accuracy | Kappa | Sensitivity | Specificity | External Overall Accuracy |
|---|---|---|---|---|---|
| **Predictor Set I** | | | | | |
| RF | 0.91 | 0.82 | 0.93 | 0.88 | 0.94 |
| SVM | 0.92 | 0.84 | 0.92 | 0.92 | 0.73 |
| LR | 0.92 | 0.83 | 0.93 | 0.90 | 0.49 |
| **Predictor Set II** | | | | | |
| RF | 0.92 | 0.84 | 0.93 | 0.90 | 0.96 |
| SVM | 0.92 | 0.84 | 0.94 | 0.90 | 0.66 |
| LR | 0.92 | 0.84 | 0.94 | 0.90 | 0.72 |

For Predictor Set II, the overall accuracy, Kappa, specificity, and sensitivity for the models are represented by nearly equal values. Only a marginal difference is noticed between Predictor Sets I and II, indicating that the climate variables contribute to the class separation. Furthermore, looking at the ROC–AUC curves seen in Figures 8 and 9, it is noticeable that the values of the models are the same for the models in both predictor sets. Additionally, in both predictor sets, the curves are above the "random guessing line", indicating that the models distinguish well between the positive and negative classes. The plotted curves show that the RF model does a slightly better job at classifying the points correctly.

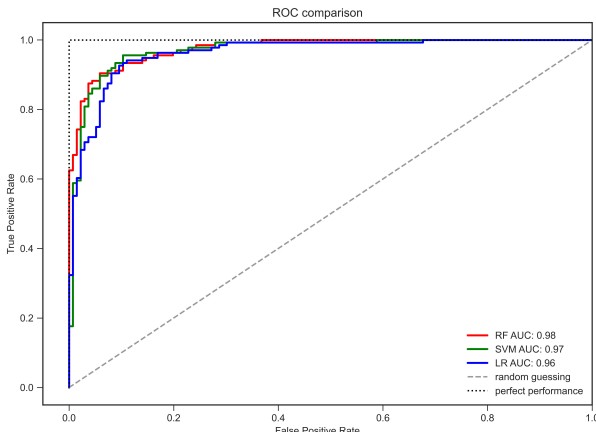

**Figure 8.** Plot of AUC–ROC curves for the ML model with Predictor Set I—without climate variables.

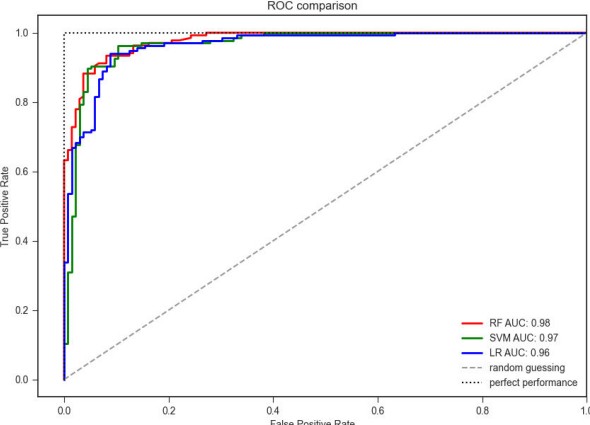

**Figure 9.** Plot of AUC–ROC curves for the ML model with Predictor Set II—with climate variables.

*Susceptibility Maps*

The susceptibility maps are visualised using equal interval classes for all the produced maps [84]. This visualisation method with equal classes makes the maps comparable. The used percentile intervals are sorted in <50%, 50–75%, 75–90%, 90–95%, and >95%, as suggested in [62].

The susceptibility maps for the three models, with Predictor Set I, are seen in Figure 10, and with Predictor Set II are seen in Figure 11. The susceptibility maps without climate variables show high and very high probabilities of landslides along the coast, and all three models indicate some risk in the northwestern area of the AOI. Generally, the models agree on which areas are susceptible to landslides; however, they differ in the assigned probability values, where SVM and LR classify a larger total area within the interval classes "high" and "very high". It is worth mentioning that SVM is not a probabilistic model, where the model results have to be converted to probabilities, so the probability output should be interpreted with caution.

The susceptibility maps with climate variables generally follow the same tendencies as the maps without climatic data. The three models generally agree on the location of the risk areas. A notable difference is the SVM model, which classifies a larger total area as being in high and very high risk when using Predictor Set II compared to Predictor set I.

The RF model is the only model that produced meaningful results with the future climatic variables. The map, seen in Figure 12, is based on the RCP8.5 scenario in years 2071–2100, and the following variables were changed according to this climate projection:

- distance_coast;
- groundwater;
- cloudburst;
- rain_max_day;
- rain_average;
- average_temp.

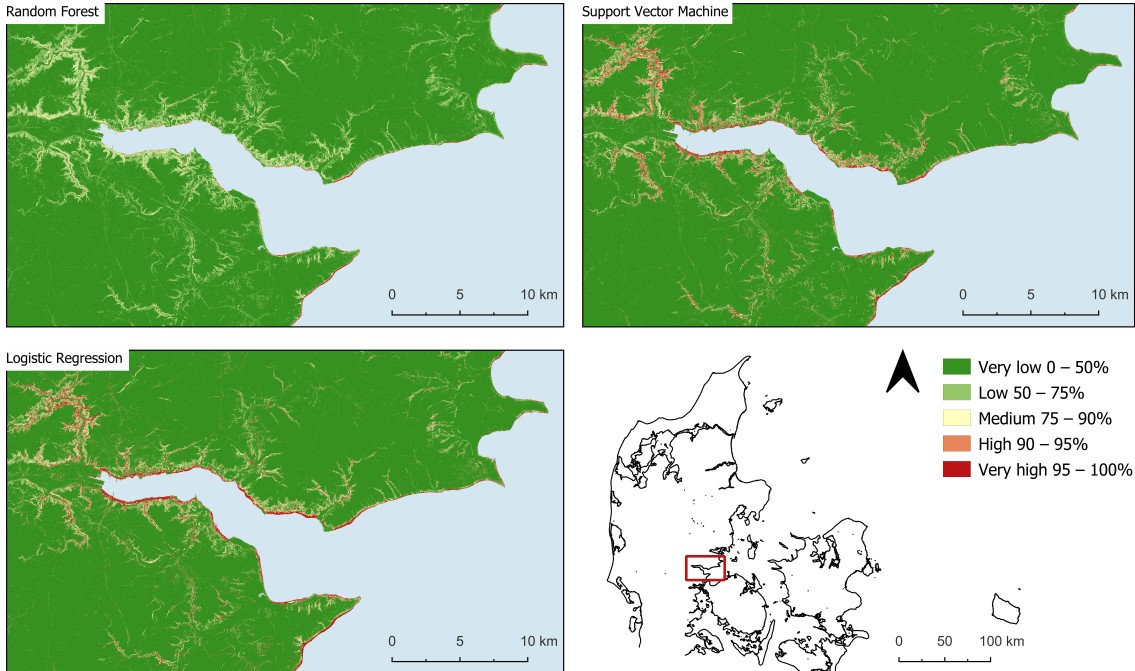

**Figure 10.** Landslide susceptibility map with Predictor Set I—without climate variables.

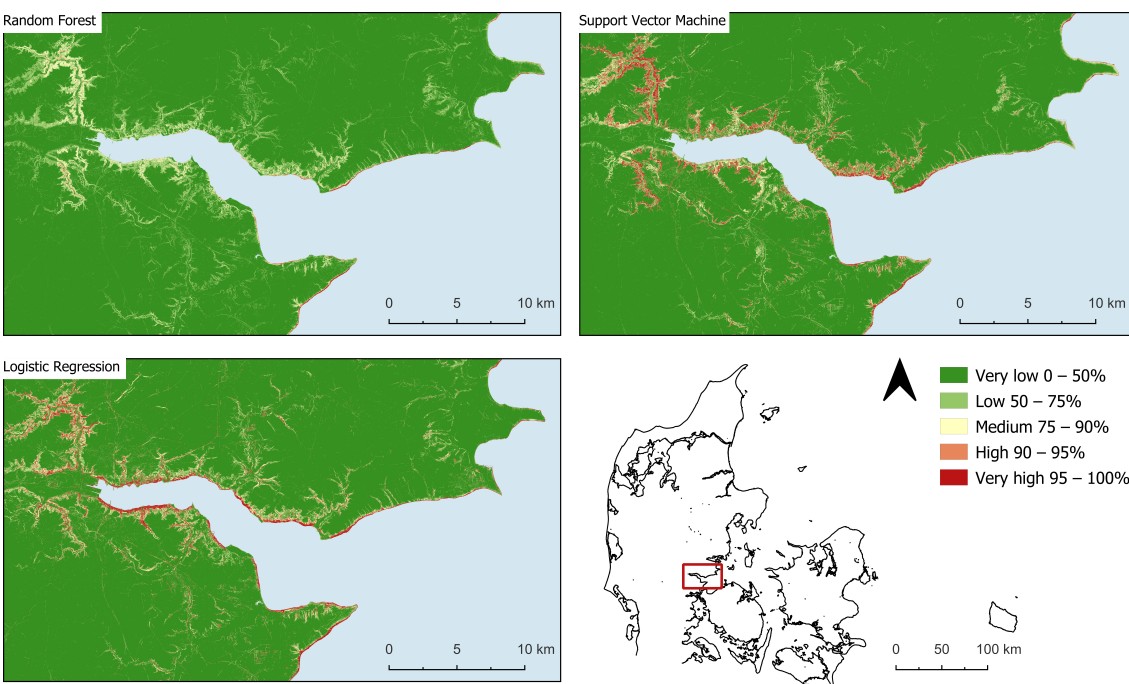

**Figure 11.** Landslide susceptibility map with Predictor Set II—with climate variables.

The future map generally shows a reduced susceptibility to landslides, without any high- or very-high-risk areas. Based on this map, the climate change would not have an increasing impact on landslide occurrences. However, it seems unlikely that climate change would have a positive impact on landslide occurrences, meaning that the use of the climatic variables in the models is questionable. A possible explanation for this is that landslides are caused by single extreme events, which are not expressed in the generalised climatic variables with a resolution of 1 km. As the landslide inventory used in the project does not contain timestamps, it was not possible to connect the landslides to certain weather events and use time-dependent predictors.

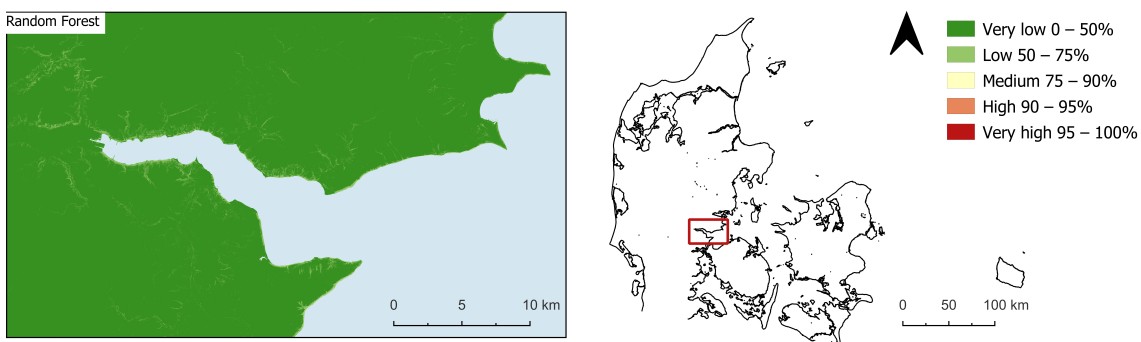

**Figure 12.** Landslide susceptibility map—future scenario RCP8.5 for 2070–2100.

## 5. Discussion

Our study showed high predictive performance with little differentiation in the three techniques applied to the AOI in the low-lying, relative flat terrain of Denmark—LR, RF, and SVM. The demonstrated predictive capacity of LR on the test data in terms of overall accuracy, specificity, sensitivity, and AUC values was higher than earlier reported by [50,85] and in line with the findings in [86]. The accuracy metrics outcome of the RF classifier was superior to the results achieved in [54], however in accordance with [54] with marginally higher AUC-values. The accuracy achieved by the SVM was higher than in the previous studies [48,87,88]. A validation on the external dataset revealed RF's overall accuracy of 94–96%, while SVM and LR exhibited inferior performance. The external validation

generally indicated that the RF is robust to new categories in the data, is able to generalise on the unknown data, and has the potential for transferability. Based on this fact and on the accuracy metrics such as overall accuracy, Kappa, and sensitivity, the recommended model for the classification task in this study is the RF. However, as the present work focused on basic ML models, a meaningful extension to it will be the calibration of more advanced algorithms, the use of hybrid modelling techniques [56,57,89], and more complex feature selection techniques [90].

By having incorporated the climate variables in this study, an attempt was made to project the impact of the future climate scenario on landslide susceptibility. The resultant map, based on the future climate data from RCP8.5 for 2071–2100, showed that climate change will reduce areas susceptible to landsliding. The results agree with the previous predictions of future susceptibility for RCP scenarios with tendencies for reduced susceptibility for some regional climate models [22]. Moreover, landslides could be caused by single extreme events, which the available reference data were not able to capture. Since the used landslide inventory has no timestamps, it is not possible to link the landslides to any particular weather events and to use time-dependent predictors [48]. Therefore, the future direction of the current work should focus on applying different reference climate data and a broader variety of climate models.

As per sensitivity analysis over the results, our findings were sensitive to the choice of input factors, the choice of machine learning algorithms, their hyperparameter settings, and the quality of our training sites. Nonetheless, the methodical approach, the incorporation of climate change variables, as well as the choice of the influencing factors in the first landslide susceptibility mapping in Denmark constitute the major contributions of this study.

## 6. Conclusions

In the present work, predictive modelling using three common ML models was performed to produce the first landslide susceptibility maps in an AOI in Denmark. The use of the three ML algorithms showed promising results for landslide susceptibility mapping in Denmark. The three classifiers (RF, SVM, and LR) trained in this study and then tested on the test data showed a high overall accuracy of 91–92%. All three models had Kappa values above 0.82 for both predictor sets, indicating that the classification was not caused by a random process. In this study, sensitivity was weighted as an important accuracy metric, as it is a measure of the proportion of the correctly classified landslides. The sensitivity for the three models was above 92%.

The RF model allows for a variable importance analysis, where it was found that the most significant variables related to landslides are DEM-derived parameters such as the elevation, TRI, and standard deviation of a slope, as well as the distance from the coast. The visual inspection of the susceptibility maps also suggests the great influence of these variables on the final result. The least important variables were found to be distances from roads, railroads, and quarries. The variable importance should be assessed in the context of the given case study and is not necessarily generalizable to other landscapes as anthropogenic activities, landscape, and geological characteristics are not identical across the globe. The DEM-derived influencing factors used in this study might enhance the footprint of the past landslides, while landslide susceptible areas, which have not yet been affected by landslides, may not exhibit such a distinct morphological expression. Thus, the landslide susceptibility mapping may require a consideration of how the terrain looked before it was altered by landslides.

At the time of the study, there is no planning for landslides in Denmark, since there is little awareness of the problem. The susceptibility maps created in this study can be used to communicate and highlight the risk of landslides for decision-makers and could potentially lay the groundwork for legislation and planning for this type of hazard in Denmark. A possible implementation of the models and resulting products in a Danish planning perspective would first of all require validation by experts within the field of landslides, since the potential implementation could have legal ramifications. Especially

the variables chosen for the models should be reviewed and analysed to ensure that only relevant variables are incorporated. At this stage, the models have only been tested on a smaller area of interest in Denmark, meaning that it could only be implemented on a regional level. If landslide planning were to be performed on a national scale, it would be necessary to extend the models.

**Author Contributions:** Conceptualization, Angelina Ageenko, Lærke Christina Hansen, Kevin Lundholm Lyng, Lars Bodum, and Jamal Jokar Arsanjani; software, Angelina Ageenko; writing—original draft, Angelina Ageenko, Lærke Christina Hansen, and Kevin Lundholm Lyng; writing—review and editing, Angelina Ageenko, Lars Bodum, and Jamal Jokar Arsanjani; supervision, Lars Bodum and Jamal Jokar Arsanjani. All authors have read and agreed to the published version of the manuscript.

**Funding:** This research received no external funding.

**Institutional Review Board Statement:** Not applicable.

**Informed Consent Statement:** Not applicable.

**Data Availability Statement:** The data, the code, and the maps of the used variables supporting the reported results are available at the online repository https://github.com/angelinkatula/Landslide-susceptibility-mapping (accessed on 17 April 2022). The national landslide inventory is publicly available from https://doi.org/10.6084/m9.figshare.16965439.v1 (accessed on 20 December 2021). The DEM, vector datasets with streams, coastline, roads, railroads, quarries, as well as depth to groundwater are available for download free of charge from https://dataforsyningen.dk/data (accessed on 05 January 2022). The geological data can be downloaded freely from https://frisbee.geus.dk/geuswebshop/ (accessed on 05-01-2022) after user registration or can be viewed through a web-GIS portal https://data.geus.dk/geusmap/ (accessed on 18 January 2022). The climate data, v2020b, are available at https://www.dmi.dk/klima-atlas/data-og-rapporter-klimaatlas/ (accessed on 10 January 2022) in the form of NetCDF-files.

**Acknowledgments:** We would like to thank Kristian Svennevig from the Geological Survey of Denmark and Greenland and Gregor Lützenburg from the Department of Geosciences and Natural Resource Management, University of Copenhagen, for their time, expert knowledge, and for providing the subset of the landslide inventory, as well as literature recommendations and valuable feedback.

**Conflicts of Interest:** The authors declare no conflict of interest.

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
