# Peer review of "Landslide Susceptibility Mapping Using Machine Learning: A Danish Case Study"

_ijgi, doi:10.3390/ijgi11060324_

Round 1

Reviewer 1 Report

This study proposes machine learning (ML)-based methods for landslide susceptibility in Denmark. The Random Forest, Support Vector Machine and Logistic Regression have been used to  classify the data samples into two classes: landslide or non-landslide. Essentially, the task of interest is modeled as a a binary classification and the ultimate output of the models is to produce susceptibility maps showing probability of landslide. Generally, the research objective is clear and the research methods incluing the data collection process and the ML model construction are appropriate. Since the studies related to ML-based landslide susceptibility in Denmark are still limited, this paper can be meaningful contribution to the body of knowledge. After reading the paper, the reviewer has the following comments:

1) Line 41: Regarding “The open data initiative in Denmark has granted wide access to actual DEMs of the entire country at 40cm spatial resolution”, please provide references or web link of the data source.

2) Introduction: The literature review is not comprehensive. Many related references have been ignored. It is suggested that the authors add a paragraph to discuss the conventional methods used for landslide susceptibility mapping [1], point out their limitations, and state the motivation for the use of ML models.

[1] A review of statistically-based landslide susceptibility models, Earth-Science Reviews, 180 (2018) 60-91.

3) Introduction: Add a paragraph to discuss the recent uses of ML in landslide susceptibility mapping, especially the works that employ the same ML models such as logistic regression [2], random forest [3,4,6], support vector machines [4]

[2] Comparison and evaluation of landslide susceptibility maps obtained from weight of evidence, logistic regression, and artificial neural network models, Natural Hazards

[3] Enhancing the accuracy of rainfall-induced landslide prediction along mountain roads with a GIS-based random forest classifier, Bulletin of Engineering Geology and the Environment, (2018)

[4] Review on landslide susceptibility mapping using support vector machines, CATENA

[5] Landslide susceptibility mapping using random forest and boosted tree models in Pyeong-Chang, Korea, Geocarto International, 33 (2018) 1000-1015.

[6] A comparative study of logistic model tree, random forest, and classification and regression tree models for spatial prediction of landslide susceptibility, CATENA, 151 (2017) 147-160.

4) In 3.3: Explain the roles of the models’ hyper-parameters

5) In 3.2: State the toolbox or library used to construct the ML models

6) Discuss the models’ performance (Random Forest, Support Vector Machine and Logistic Regression) in light of the findings of the previous works (e.g. ref. [2,3,5,6] )

7) According to Table 2, there is no significant difference between the result of the Predictor set I and that of the Predictor set II. Please provide a paragraph to discuss or explain this phenomenon.

8) According to Table 2, RF has very good performance in both internal and external datasets. However, SVM and LR do not. Does this mean that RF has generalize the collected data well and SVM and LR has been suffered from overfitting? Alternatively, are there distinctive features in the external dataset that cause the poor performances of SVM and LR. Please provide a paragraph to discuss or explain this phenomenon.

9) In 5.1: future extensions of the current work should take into account the applications of more advanced ML models and feature selection algorithms.

Reviewer 2 Report

Landslide susceptibility mapping using machine learning: A Danish case study

This subject addressed is within the scope of the journal. Manuscript in the present version contains several problems. Most important question is the novelty of the paper.

  1. Need to revise the abstract because it does not reflect the work you have done actually
  2. The words already resent in the title of the manuscript not needed to mention again in keywords section.
  3. In this study authors have used several categorical variables like geology, geomorphology, etc. How the authors standardized the categorical variable not clear.
  4. Why not tried hybrid models for comparison? For example, LSTM-ALO, DENFIS, GMDH, LSSVM-GSA recently used in literature about modeling. Should add these models recent literature and also explain why not adopted those advanced version?
  5. To quickly catch your contribution by the readers, it would be better to highlight major difficulties and challenges, and your original achievements to overcome them, in a clearer way in abstract and introduction.
  6. There is a serious concern regarding the novelty of this work. What new has been proposed?
  7. The discussion section in the present form is relatively weak and should be strengthened with more details and justifications. Also need compare the result with other published work.
  8. Why not draw scatter, Taylor and violin plots to compare the results?
  9. Authors need to add the sensitivity analysis for pointing out the importance of the individual variable.

Reviewer 3 Report

The present study generated regional landslide susceptibility map by using three different machine learning models. The methods and modelling procedure have no innovation, but climate change was considered which makes me happy. I suggest a major revision, here you can find my concerns:

1) Introduction section: some up-to-date work are missing:

regarding the machine learning for landslide susceptibility:

Zizheng Guo, Yu Shi, Faming Huang, Xuanmei Fan, Jinsong Huang. Landslide susceptibility zonation method based on C5.0 decision tree and K-means cluster algorithms to improve the efficiency of risk management. Geoscience Frontiers, 2021.

2) As we can see from the Figure 1, most landslides are near from the sea, and this is why the variable of distance to coast has large importance. Do you know why it is like this? What is the main mechanism of the landslides in the inventory? Could you try to analyze it? Just to have an overall idea on the landslides in the study area.  

3) L406-426. I don’t understand why you described these contents here. The section 3.4 is accuracy assessment, but this is not related with accuracy.

4) Figure 4 is not clear. I can’t see the details well, such as legends and axis. Please improve it. This is also the case for Figure 5.

5) Where is your conclusion section? I only find the Discussion. Please add it.

Round 2

Reviewer 1 Report

I have no further comment.

Author Response

No further comments.

Reviewer 2 Report

Now paper can be accepted for publication. 

Author Response

No further comments.

Reviewer 3 Report

Dear authors,

Most of my comments have been addressed. However, I noticed a drawback in the section Introduction. You mentioned that one of big objectives in this study is that "Can the impact of changing climate on landslide susceptibility be modelled for the future climate scenarios?" (L131-132). However, the Introduction only focused on landslide susceptibility mapping tecniques. The description on the impact of climate change on landslide susceptibility mapping is MISSING. In my opinion, this part is necessary because without it, your Introduction has no difference with other ordinary studies that discuss the comparison of model performace. Please keep in mind that the biggest novel thing of this study is that you are considering climate change. So please highlight it in the Introduction. Some relative literature have discussed it. You should list and discuss them. You may reduce the sub-section 1.1 because the contents in it are really common.
